# The Role of Autophagy in Anti-Cancer and Health Promoting Effects of Cordycepin

**DOI:** 10.3390/molecules26164954

**Published:** 2021-08-16

**Authors:** Yu-Ying Chen, Chun-Hsien Chen, Wei-Chen Lin, Chih-Wei Tung, Yung-Chia Chen, Shang-Hsun Yang, Bu-Miin Huang, Rong-Jane Chen

**Affiliations:** 1Department of Environmental and Occupational Health, College of Medicine, National Cheng Kung University, Tainan 70101, Taiwan; 101312123@gms.tcu.edu.tw; 2Institute of Basic Medical Sciences, College of Medicine, National Cheng Kung University, Tainan 70101, Taiwan; greatwall91983@gmail.com; 3Department of Parasitology, College of Medicine, National Cheng Kung University, Tainan 70101, Taiwan; wcnikelin@mail.ncku.edu.tw; 4Department of Microbiology and Immunology, College of Medicine, National Cheng Kung University, Tainan 70101, Taiwan; 5Department of Physiology, College of Medicine, National Cheng Kung University, Tainan 70101, Taiwan; b90607021@ntu.edu.tw; 6Department of Anatomy, School of Medicine, Kaohsiung Medical University, Kaohsiung 80708, Taiwan; yungchia@kmu.edu.tw; 7Graduate Institute of Medicine, College of Medicine, Kaohsiung Medical University, Kaohsiung 80708, Taiwan; 8Department of Medical Research, China Medical University Hospital, China Medical University, Taichung 40402, Taiwan; 9Department of Cell Biology and Anatomy, College of Medicine, National Cheng Kung University, Tainan 70101, Taiwan; 10Department of Food Safety/Hygiene and Risk Management, College of Medicine, National Cheng Kung University, Tainan 70101, Taiwan

**Keywords:** cordycepin, autophagy, anti-cancer, health-promoting effects, neuron, kidney, immune systems

## Abstract

Cordycepin is an adenosine derivative isolated from *Cordyceps sinensis*, which has been used as an herbal complementary and alternative medicine with various biological activities. The general anti-cancer mechanisms of cordycepin are regulated by the adenosine A3 receptor, epidermal growth factor receptor (EGFR), mitogen-activated protein kinases (MAPKs), and glycogen synthase kinase (GSK)-3β, leading to cell cycle arrest or apoptosis. Notably, cordycepin also induces autophagy to trigger cell death, inhibits tumor metastasis, and modulates the immune system. Since the dysregulation of autophagy is associated with cancers and neuron, immune, and kidney diseases, cordycepin is considered an alternative treatment because of the involvement of cordycepin in autophagic signaling. However, the profound mechanism of autophagy induction by cordycepin has never been reviewed in detail. Therefore, in this article, we reviewed the anti-cancer and health-promoting effects of cordycepin in the neurons, kidneys, and the immune system through diverse mechanisms, including autophagy induction. We also suggest that formulation changes for cordycepin could enhance its bioactivity and bioavailability and lower its toxicity for future applications. A comprehensive understanding of the autophagy mechanism would provide novel mechanistic insight into the anti-cancer and health-promoting effects of cordycepin.

## 1. Introduction

Cordycepin is an adenosine derivative isolated from *Cordyceps sinensis*, which is a fungal parasite of the larvae of Lepidoptera, especially the ghost moth [1]. The fungus attacks and devours the caterpillars in late autumn; after killing the caterpillar, the fruiting body protrudes from its head in the early summer of the following year. Thus, it is called “winter-worm summer-plant” in Chinese [2,3,4]. *Cordyceps sinensis* has been used extensively as an herbal complementary and alternative medicine drug to treat different illnesses in oriental society on the basis of scientific studies showing that it has multiple pharmacological activities, including immune response modulations [5,6], blood pressure reduction [7,8], hepatic energy metabolism and blood flow regulation [9], body bioenergy improvement [10], hormone secretion augmentation [11,12,13], apoptosis induction [14,15], tumor growth inhibition [15,16,17], etc. 

However, *Cordyceps sinensis* contains numerous substances, and the specific pure component responsible for the pharmacological activities remains elusive. Cordycepin is the major component in *Cordyceps sinensis* [18,19], and investigations of the positive effects of cordycepin on different diseases and health issues have boomed recently, with reviews showing its activities with anti-inflammatory effects [20], immunomodulatory effects [21], anti-diabetic effects [22,23], anti-hyperlipidemic effects [22], antioxidant effects [18,24], prevention in cardiovascular diseases [22], anti-cancer effects [25], etc.

Various studies on the anti-cancer effects of cordycepin has been described [16,26,27,28], and the mechanisms through which cordycepin kills tumor cells and suppresses tumor growth have been studied and summarized to show that cordycepin may induce tumor cell death through cysteine–aspartic proteases (caspases), mitogen-activated protein kinase (MAPK), and glycogen synthase kinase (GSK)-3β pathways mediated by putative adenosine receptors, death receptors, and/or epidermal growth factor receptors (EGFR) [16,27,29]. Especially, the stimulation of the adenosine A3 receptor activates synthase kinase GSK-3β and suppresses cyclin D1 [16,27] and then induces cell cycle arrest and DNA damage to further regulate the tumor microenvironment and target cancer stem cells [19,28,30].

In addition, cordycepin regulates phosphoinositide 3-kinase/protein kinase B (PI3K/AKT) signaling pathways and suppresses the expression of cyclin-dependent kinase 2 (Cdk-2), extracellular signal-regulated kinase 1/2 (ERK1/2), Rb/E2F1, and fibroblast growth factor receptor 1-4 (FGFR 1-4) to regulate the cell cycle, further reducing the growth of testicular tumors, gastric cancer cells, and cervical cancer cells [31,32,33,34]. Moreover, cordycepin has been reported to activate caspase cascades and elevate intracellular reactive oxygen species (ROS) levels to eliminate human tongue cancer cells, testicular tumor cells, and human OEC-M1 oral cancer cells [29,33,35]. Furthermore, cordycepin also regulates different signaling proteins, such as hedgehog, glioblastoma proteins (GLI), DNA-dependent protein kinase (DNA-PK), and ERK, to induce the apoptosis of cancer cells [36,37] and activates centrosome amplification and aberrant mitosis to disrupt human gestational choriocarcinoma cells [38]. All these studies elucidate that cordycepin may activate different cell signaling pathways to induce cell death and/or apoptosis among different tumor cell types.

Interestingly, cordycepin also induces autophagy to trigger cell death, inhibits tumor metastasis, and modulates the immune system [24,28,38]. However, the relationship between cordycepin and autophagy-related anti-cancer and health-promoting effects has rarely been explored. Thus, we will briefly review and highlight the potential platforms for future research regarding autophagy that are relevant to the anti-cancer and health-promoting effects of cordycepin. Regarding the health-promoting effects of cordycepin, the topics of kidney diseases, neurodegenerative diseases, and microbiota with immune system modulation related to autophagy will be revealed in this article, as the above topics are captivating and important but have not been satisfactorily investigated and depicted.

## 2. The Role of Autophagy in the Anti-Cancer Effects of Cordycepin

Autophagy is a catabolic mechanism that is evolutionarily conserved from yeast to mammals. The autophagy pathway, first described by Christian De Duve in 1963 [39], is an ubiquitous process that involves the degradation of cytoplasmic components and cytoplasm organelles through the lysosomal pathway and is distinct from other degradative pathways, such as proteasomal degradation [40]. When cells encounter DNA damage, hypoxia, starvation, or chemotherapy drugs, AMP-activated protein kinase (AMPK) is activated, which can drive autophagy. Similarly, the deprivation of growth factors or amino acids could lead to the inhibition of the target of rapamycin complex 1 (TORC1), which is a repressor of conventional autophagy [41]. Autophagy plays a major role in the degradation of damaged organelles and old proteins and in the maintenance of cellular homeostasis (Figure 1) [42]. The dysregulation of autophagy is associated with many types of diseases, including cancers, neurodegenerative diseases, kidney diseases, immune dysfunction diseases, and so on [43]. In cancer biology, autophagy plays dual roles in tumor promotion and suppression, which contributes to cancer cell development and/or proliferation [44].

In the tumor promotion role, autophagy acts to promote tumor growth and survival in advanced cancers [45]. In cancer cells, autophagy provides for the high metabolic and energetic demands of proliferating tumors by recycling intracellular components to supply metabolic substrates [46]. One study showed that a high level of autophagy occurs in RAS-activating mutated cells, and cell survival is dependent on autophagy during nutrient starvation [47]. Therefore, autophagy contributes to tumor cell survival by enhancing stress tolerance and supplying nutrients to meet the metabolic demands of tumors, and the inhibition of autophagy or knockdown of autophagy genes can result in tumor cell death [44].

On the other hands, the late stage of tumor progression is usually subject to environmental stress, including limited angiogenesis, nutrient deprivation, and hypoxia. In this stage, autophagy contributes to the survival and growth of the established tumors and promotes the aggressiveness of cancers by facilitating metastasis [44,48]. Previous studies have reported that the downregulation of Beclin-1 is observed in human breast, prostate, and ovarian cancers [44]. Other studies have shown that the deficiency of other autophagic regulators, such as the Autophagy-Related genes (*ATG*) family, is related to carcinogenesis [48]. Numerous studies have shown that other *ATG* genes, including *ATG2B*, *ATG5*, *ATG9B*, *ATG12*, and *ATG16L1*, are also associated with human different cancers. The mutations with mononucleotide repeats have been found in *ATG2B*, *ATG5*, *ATG9B*, and *ATG12* genes in gastric and colorectal cancers, which may be involved in cancer development by dysregulating autophagy [48].

Taken together, autophagy has dual roles in the progression or inhibition of cancers. Autophagy acts as a tumor suppressor early in the progression, whereas it acts as a cancer promotor later in tumor maintenance and cancer therapy resistance. In early carcinogenesis, autophagy plays a role in the survival and quality-control mechanism, contributes to normal cell physiology metabolism, provides biological materials and energy in response to stress, and regulates dynamic degradation and quality-control mechanisms to eliminate damaged proteins and organelles, thus preventing tumor initiation. Therefore, autophagy can be used as an effective interventional strategy for cancer prevention and therapy in each stage of cancer, limiting tumor development and progression. To understand the in-depth mechanisms of cordycepin in cancer prevention and therapy, we speculate that the autophagy mechanism could be an important target of cordycepin worthy of further investigation.

However, studies on cordycepin and autophagy related to cancer cell death are infrequent, and this relationship has only been investigated in some reports, which illustrated that cordycepin could induce cell death through autophagy in breast cancer cells [49], prostate carcinoma [21], neuroblastoma [50], non-small lung cancer [51], brain cancer [52], and ovarian cancer cells [53]. Among the few studies that observed cordycepin-induced cell death through autophagy in different cancer cells [21,49,51,52,53], mechanism descriptions were brief and described that cordycepin could trigger an increase in the LC3-II to LC3-I protein ratio, inhibition of the β-catenin pathway, and/or the suppression of the mechanistic target of rapamycin (mTOR) signaling pathway (Figure 1). Thus, profound mechanism investigations regarding the anti-cancer effect of cordycepin via autophagy should be further investigated, which could highlight the optimal function of cordycepin as a decent chemotherapy drug.

### 2.1. Cordycepin-Induced Cancer Cell Death through Autophagy Induction

As described above, several studies have shown that cordycepin could induce cancer cell death through autophagy [21,49,51,52,53]. In fact, we found some interesting observations regarding the role of autophagy with a deeper examination of the mechanism of cordycepin in regulating cell death in MA-10 mouse Leydig tumor cells [29], oral squamous carcinoma [54], and human gestational choriocarcinoma cells [38]. In MA-10 cells, cordycepin inactivates PI3K/AKT/mTOR signaling pathways and upregulates LC3-II expression to regulate cell death [29]. In oral squamous carcinoma cells, cordycepin with irradiation induces the S-phase and prolongs G2/M arrest and the upregulation of the autophagy pathway to elicit cell death [54]. In human gestational choriocarcinoma JAR cells, cordycepin activates DNA-PK and ERK to induce centrosome amplification and aberrant mitosis, which disrupts centrosome homeostasis to induce autophagy and trigger cell death [38]. Cordycepin-stimulated autophagy via the suppression of the mTOR signaling pathway in lung cancer cells has been reported; the suppression of autophagy also elevates the expression level of cellular FLICE-like inhibitory protein (c-FLIP), indicating that cordycepin-triggered autophagy promotes the degradation of c-FLIP to induce apoptosis through autophagy-mediated downregulation of c-FLIP in human non-small cell lung cancer (NSCLC) cells [51]. In addition, cordycepin also inhibits the ERK/Slug signaling pathway through the activation of GSK-3β, which upregulates Bax and results in the apoptosis of lung cancer cells [55].

Although some detailed mechanisms have been demonstrated in our studies, as shown above, further in vitro experiments with more profound mechanism investigations and animal in vivo studies should be conducted to correlate the in vitro and in vivo results to highlight the efficacy of cordycepin to activate autophagy for cancer treatments.

### 2.2. The Effects of Combination Therapy of Cordycepin and Anti-Cancer Therapy

We also found that cordycepin enhances the anti-cancer activity of cisplatin and/or paclitaxel in two different human head and neck tumor cells [56,57] and in MA-10 mouse Leydig tumor cells [28], respectively. Regarding the mechanisms of cordycepin enhancing cisplatin anti-cancer activity, we demonstrated the activation of c-Jun N-terminal kinase (JNK) and caspase pathways in head and neck tumor cells [56,57] and the activation of JNK, p38, p53, and caspase pathways in MA-10 cells [58]. In these studies, an exploration of the autophagy phenomenon was not executed and should be conducted soon. In another study, however, we did observe that cordycepin enhances radiosensitivity in oral squamous carcinoma cells with the upregulation of *ATG*5 and p21 proteins and cell cycle arrest in an autophagy cascade-dependent manner [54].

In fact, studies of cordycepin combined with anti-cancer drugs inducing tumor cell death that investigate the mechanisms are seldom conducted. Only two reports have shown that cordycepin combined with anti-cancer drugs induces apoptosis in NSCLCs by activating the AMPK signaling pathway [59] and in human glioma cells by activating AMPK and inhibiting the AKT signaling pathway [60]. There is a report showing that cordycepin also enhances radiosensitivity in oral cancer cells associated with the modulation of DNA damage repair [61]. However, these three papers did not examine the autophagy issue. Thus, deeper in vitro mechanism investigations and animal in vivo studies related to autophagy should be conducted to increase the merits of combination therapy of cordycepin and anti-cancer drugs and/or radiation.

## 3. The Role of Autophagy in the Health-Promoting Effects of Cordycepin

As described above, autophagy is induced by various cellular stressors [62]. Notably, autophagy can also act as a double-edged sword depending on cell and disease conditions [63,64,65]. In this review, we further discuss the roles of autophagy in the regulation of diseases, particularly those of the kidneys, neurons, and immune system. The therapeutic potential and challenges of targeting autophagy using cordycepin for the prevention and treatment of these diseases are also discussed.

### 3.1. The Role of Autophagy in Kidney Disease

Autophagy has an important role in kidney development and the reduction of kidney diseases (Figure 2). Basal autophagy is essential for the maintenance of kidney homeostasis, structure, and function. However, dysregulated autophagy, such as autophagy dysfunction or overactivation, contributes to the pathogenesis of acute kidney injury (AKI), chronic kidney disease (CKD), and polycystic kidney disease (Figure 2). In the kidney, podocytes show high levels of autophagy and have a normal autophagic flux [66]. Mice with knockdown of *ATG5* in progenitor epithelial cells of podocytes, parietal epithelial cells, the proximal tubule, the loop of Henle, and distal tubule cells showed severe glomerular and tubular injury and progressive renal dysfunction that was similar to that of human CKD [67]. Moreover, mice with autophagy dysfunctions develop severe glomerulosclerosis and proteinuria, highlighting the importance of autophagic flux and regulation signaling, including TOR–autophagy spatial coupling compartments (TASCCs), mTORC1, and AMPK, for the maintenance of podocyte homeostasis [66,67,68,69].

In contrast to podocytes, renal tubular epithelial cells (RTEC) show lower baseline levels of autophagy; however, knockout of ATG5 or ATG7 also results in tubular injury during development [62]. Similarly, inhibition of autophagy by inhibitors or gene silencing aggravates nephropathy induced by chemicals, drugs, or food components [70,71,72]. For instance, autophagy was induced in the early stage of glucose or streptozotocin (STZ) treated-diabetic mice, while autophagy was reduced at later time points [73]. However, knockdown of ATG5 aggravates STZ-induced kidney disease [73]. Similar results were found in kidney biopsy samples from diabetic kidney disease patients with massive proteinuria, which showed reduced autophagy levels in podocytes [74]. Moreover, clinical trials that tested autophagy inducers, such as metformin, showed significant benefits in delaying the onset and severity of proteinuria in diabetic kidney diseases [75].

The beneficial effect of autophagy was further confirmed by Wang et al., who used a high adenine-induced urate nephropathy model in which NOD-like receptor (NLR) family pyrin domain containing 3 (NLRP3) inflammasome activation and renal fibrosis were detected in the kidney. Treatment with an autophagy inducer, pterostilbene, significantly reduced NLRP3 inflammasome activation and renal fibrosis after transforming growth factor beta (TGF-β) stimulation [71]. Several studies using gene knockout (KO) animals or cell lines have confirmed the protective role of autophagy in renal fibrosis. *ATG5* KO mice show more susceptibility to unilateral ureteral obstruction (UUO)-induced kidney fibrosis, concomitant with cellular senescence, DNA damage, and cell cycle arrest [76,77]. *ATG7* KO mice with impaired autophagy also show accelerated renal tubulointerstitial fibrosis and the increased expression of TGF-β, interleukin-1β (IL-1β), NLRP3, endoplasmic reticulum (ER) stress, and mitochondria damage in the kidney of the UUO-treatment model [67,78]. Taken together, these data suggest that autophagy plays a protective role in kidney development and renal diseases, while inhibited autophagy in the context of injury stimuli aggravates renal damage. Accordingly, induction of autophagy could offer a therapeutic benefit in the treatment of renal diseases.

By contrast, many studies have also suggested that excessive autophagy is related to kidney diseases in various stress models [79,80,81,82,83]. After AKI, the injured tissue should be repaired after tubular cell proliferation, migration, and differentiation [84,85]. However, when the repair of severe AKI is incomplete, the damage is transformed into the CKD later [69]. Fibrosis is a hallmark of maladaptive repair in the transition from AKI to CKD and is also a characteristic of CKD in the late stage [86]. Autophagy dysfunction or persistent autophagy impairs tubular cell proliferation, leading to fibrosis and aggravated CKD. For instance, WNT family member 1 (WNT1)-induced signaling pathway protein-1 (WISP-1), Myc, CCAAT-enhancer-binding protein (C/EBP), and protein kinase C alpha (PKCα) can accelerate the development of renal fibrosis by increasing autophagic flux in the UUO-induced mouse model and TGF-β-stimulated tubular epithelial cells [80,81,82,83]. In fact, we previously reported that kidney damage along with fibrosis is observed in mice exposed to food contaminant 3-monochloropropane-1,2-diol (3-MCPD) and glycidol alone or in combination [63]. Autophagic cell death, necroptosis, and pyroptosis occur simultaneously in damaged kidney tissues and tubular cell lines as well [63]. Therefore, we proposed that an interaction existed between autophagic cell death and other cell modalities, which thereafter contributed to kidney toxicity by 3-MCPD and glycidol coexposure [63].

In accordance with our study, a previous study indicated that autophagy has pro-fibrotic effects and is involved in the activation of kidney fibroblasts in UUO mice and cultured kidney fibroblast cells after exposure to TGF-β1 [82]. Autophagy impairment is also detected in high glucose-treated proximal tubular epithelial cells (PTECs), the kidneys of diabetic mice, and kidney biopsy samples from patients with diabetic kidney diseases (DKD) [87]. Collectively, these findings suggest that persistent activation or impairment of autophagy might induce tubular atrophy and promote kidney fibrosis, thereby aggravating kidney diseases. The mechanisms involved in this effect are very diverse, and readers are referred to a comprehensive review with a detailed discussion of the mechanisms [69]. Although these studies have yielded many important advances in the understanding of the role of autophagy in kidney diseases, data on the underlying mechanisms of autophagy in kidney homeostasis and diseases are still needed from further investigations.

### 3.2. The Role of Autophagy in Neurodegenerative Diseases

Neurodegenerative diseases are common disorders in modern society and have affected millions of patients worldwide. These diseases include Alzheimer’s disease (AD), Parkinson’s disease (PD), Huntington’s disease (HD), etc., and display neurodegeneration causing neuronal death, further leading to clinic symptoms of motor or cognitive dysfunction [88,89,90,91]. Most interestingly, these diseases share certain similar cellular and pathological characteristics, such as disruption of protein degradation systems and accumulations of disease proteins. In AD, PD, and HD, two important protein degradation systems, the autophagy system and ubiquitin proteasome system (UPS), have been reported to be dysfunctional during the progression of diseases and result in the formation of disease protein aggregates, including amyloid precursor proteins (APP) in AD, α-synuclein in PD, and Huntingtin in HD (Figure 3). These aggregates form critical neuropathological features, such as amyloid β (Aβ) plaques in AD; Lewy bodies in PD; and nuclear, intranuclear, and neuropil aggregates in HD, and then disturb other cellular functions in the central nerve system (CNS) [92,93,94]. As a result, improvements in these two protein degradation systems, especially autophagy, are considered a potential direction for therapy for these neurodegenerative diseases.

In AD, deficits in *ATG*s, such as ATG5, ATG7, and Beclin-1 [95,96], along with autophagy-related processes, such as mitophagy and mTOR activation [97,98,99], have been reported in AD models. These deficits are observed in neurons, astrocytes, and microglial cells, suggesting that abnormal protein degradation occurs throughout the whole CNS. Tau and APP proteins are two critical components inside Aβ plaques, and these two proteins can be degraded via autophagy in normal people. However, because of the deficits in autophagy in AD, patients show accumulations of Aβ plaques in their brains. As a result, treatments that target reducing Tau and APP through autophagy have emerged and have been shown to alleviate the pathology of AD models [95,100,101].

In PD, autophagy-related proteins and autophagy-related processes have displayed impairments in cellular functions as well, and the mutations in autophagy-related genes, such as *ATG5* and *ATG12*, have been found to serve as risk factors affecting the disease progression in PD patients [102,103,104]. Because of the deficits of autophagy, α-synuclein, ubiquitin, tau protein, etc., assemble to form Lewy bodies and eventually lead to neuropathological features and clinic symptoms. To enhance the functions of autophagy, several treatments, such as prolyl oligopeptidase [105], piperlongumine [106], and autophagy enhancer-99 (AUTEN-99) [107], have been applied in different PD models, and these drugs have been shown to activate autophagy to alleviate PD phenotypes. These results highly suggest that the enhancement of autophagy is a promising strategy to alleviate the progression of PD.

HD is a polyglutamine disease caused by the translation of abnormally expanded cytosine-adenine-guanine (CAG) trinucleotides in the disease-causing gene *Huntingtin*. The abnormally expanded polyglutamines are misfolded and then form nuclear, intranuclear, and neuropil aggregates to disrupt cellular functions [108,109,110]. These polyglutamine expansions have been shown to regulate Beclin-1 to influence autophagy in different polyglutamine diseases [111], and the abnormal expansion of polyglutamines has been reported to disrupt Beclin-1 and autophagy to cause aggregates in HD [112,113]. In addition to Beclin-1, the abnormal expression or location of autophagy-related proteins, such as LC3 and sequestosome 1 (SQSTM1/p62), have been reported in different HD models [114,115]. These results show the deficits of the autophagy system in HD and suggest that therapy targeting autophagy is a potential direction for this disease. Indeed, several drugs, such as minoxidil, clonidine, phenoxazine, everolimus, memantine, and clemastine, have been observed in different HD models to induce the autophagy system and have been shown to improve or alleviate HD phenotypes [116,117,118,119]. Taking these results for AD, PD, and HD together, autophagy not only plays an important role in cleaning disease-causing proteins inside cells but also serves as a potential direction to treat neurodegenerative diseases.

### 3.3. The Role of Autophagy in Immune Systems

Many people worldwide suffer from immune disorders, such as auto-immune diseases and inflammatory syndromes. It has been reported that autophagy is involved in most cellular stress-response pathways, such as immunity regulation, which includes the adjustment of various pathogen-recognition receptors as well as the inflammasome (Figure 4) [120,121]. In the innate immune system, *ATG*16L1, interferon-inducible GTPase, autophagy-associated proteins, and nucleotide-binding oligomerization domain 2 are important in antibacterial responses and Crohn’s disease [122,123]. Autophagy and autophagy-associated proteins are also required for antigen-presenting signaling and lymphocyte development in the adaptive immune system [124,125,126]. Therefore, autophagy plays a crucial role in the regulation of the innate and adaptive immune responses.

Several studies have indicated that autophagy participates in the maintenance of the innate immune response by regulating the clearance of microbial pathogens in macrophages and dendritic cells [127,128,129]. One way that this occurs is through xenophagy, a unique type of selective autophagy that targets pathogens and eliminates pathogens by engulfing them in cytosol autophagosomes [130,131]. Microtubule-associated protein LC3 is involved in another way, through LC3-associated phagocytosis (LAP), to remove pathogens by enclosing them in single-membrane phagosomes [132]. In addition, during phagocytosis, the engagement of toll-like receptors recruits *ATG*12 and LC3 to the phagosome via the NADPH oxidase 2-dependent production of ROS [127].

LAP, which is initiated by toll-like receptor agonists and immune complexes, plays a vital role in removing extracellular particles, including pathogens in phagocytes, particularly macrophages [128,133,134]. LAP is also instrumental in dead cell clearance, which requires the phosphatidylserine (PtdSer) receptor T-cell immunoglobulin mucin-4 (TIM4) to trigger LC3 recruitment to the phagosome [135,136]. A lack of LAP results in a decreased capacity to clear apoptotic cells and the development of immune disorders, such as lupus-like autoimmune diseases, in mice [137]. Thus, the autophagy pathway may reduce inflammation via its role in apoptotic cell clearance.

Another way that autophagy proteins regulate inflammatory signaling is by adjusting the level of inflammasomes. Inflammasomes containing NLRs, the adaptor protein Apoptosis-associated speck-like protein containing a caspase recruitment domain (ASC), and pro-caspase 1 have been well-studied in terms of promoting the production of pro-inflammatory cytokines, including IL-1β and IL-18, in the immune cells of the innate immune system [138]. Mice lacking *ATG*16L1 macrophages produced higher levels of IL-1β and IL-18 secretions after lipopolysaccharide (LPS) induction, which may contribute to increased dextran sulfate sodium-induced colitis [139].

Autophagy is also involved in adaptive immunity, including the development of the immune system and antigen presentation [140]. Antigen-presenting cells (APCs), such as dendritic cells, macrophages, and B cells, connect the innate and adaptive immune responses [141]. Autophagy is necessary for the presentation process of major histocompatibility complex proteins (MHC) in APCs [142]. UPS and autophagy trigger proper pathogen digestion to support MHC class II (MHC-II) antigen presentation [143]. It has been reported that starvation induces autophagy to enhance peptide presentation [124]. Mice lacking *ATG*5 in dendritic cells show defective T cell responses after herpes simplex virus and listeria infection [124,126].

Autophagy has also been linked to cytokine production in dendritic cells, such as IL-12p40 and IL-6. In addition, the inhibition of autophagy in dendritic cells significantly reduces cytokine production by CD4+ T cells in the antigen-stimulated condition [144]. The specific deletion of *ATG*5 in dendritic cells appears to delay the fusion of phagosomes to lysosomes and inhibits the process of presenting phagocytosed antigens containing toll-like receptor ligands on MHC-II molecules [145,146,147]. Several autophagy proteins also regulate B cell survival and development. Yuan et al. found that IL-17-elevated autophagy enhances UPS activity and increases B cell anti-apoptotic ability by stimulating Beclin-1 and p62 expression [148]. In contrast to pre-B and mature B1a B cell survival, a correlation analysis showed that the expression levels of apoptotic proteins, including Beclin-1, LC3, and p62, are positively correlated with the systemic lupus erythematosus disease activity index, which can evaluate the disease activity of systemic lupus erythematosus [149,150]. Collectively, dysregulated autophagy can result in disordered immune cell homeostasis and immune cell dysfunction, which contribute to immunodeficiency and/or auto-immune diseases.

### 3.4. Potential Health-Promoting Effects of Cordycepin via Autophagy Induction

Several previous review articles have illustrated the health-promoting effects of cordycepin [20,22,27,151]. A previous study also showed that cordycepin activates AMPK to block the activity of mTORC1 and mTORC2 complexes by unknown mechanisms [19]. It is known that AMPK stimulation and mTOR inhibition act as initial signals to activate autophagy. Thus, we propose that cordycepin may regulate autophagy to initiate important health-promoting effects. Accordingly, we focus on summarizing the beneficial effects of cordycepin in targeting autophagy-regulated diseases, including diseases of the kidneys, neurons, and immune system (Figure 5). As described above, autophagy is an important mechanism in regulating kidney homeostasis and diseases. Cao et al. illustrated that cordycepin suppressed cell apoptosis and renal fibrosis and rescued cell autophagy in a diabetic nephropathy (DN) rat model [152]. However, this study lacked an explanation of the detailed molecular mechanisms related to autophagy induction by cordycepin. Yong et al. also reported that cordycepin has the potential to produce an anti-hyperuricemic effect in mice through the downregulation of uric acid transporter 1 (URAT1) [153]. Moreover, cordycepin was reported to improve the progression of CKD by affecting the Toll-like receptor 4 (TLR 4)/nuclear factor-kappa B (NF-κB) redox signaling pathways [154]. In UUO models, the metabolite of cordycepin was reported to interfere with TGF-β and bone morphogenetic protein-4 (BMP-4) signaling by downregulating Smads in vitro and in vivo [155]. These studies indicate that cordycepin could prevent kidney damage in various models by modulating inflammation and autophagy-related pathways. Therefore, the molecular mechanisms of autophagy induced by cordycepin in ameliorating renal damage need urgent investigation.

Moreover, cordycepin has been reported to regulate neuronal functions [156,157,158,159], and further, to provide beneficial functions in several neuronal diseases, such as AD, PD, Machado–Joseph disease, and ischemia [160,161,162,163]. The protective mechanisms of cordycepin have been shown to regulate 5-hydroxymethylcytosine levels, TLR/NF-κB signaling, AMPK activity, acetylcholinesterase activity, etc. to offer anti-inflammation, anti-oxidation, and improvements in mitochondrial and autophagic functions [37,151,161,164]. Since neurodegenerative diseases display multiple pathological characteristics, including inflammation, oxidative stress, dysfunction of mitochondria, and abnormalities in the UPS and autophagy system simultaneously, cordycepin, targeting multiple cellular functions, is considered to be an advantageous treatment to apply to these diseases. Most interestingly, cordycepin was shown to activate the autophagy system and reduce ubiquitin aggregates, which is considered a critical characteristic in several neurodegenerative diseases [162]. Although there are limited references addressing the autophagy functions of cordycepin in neurodegenerative diseases, cordycepin does induce autophagy in other diseases, as described in this review, highly suggesting the potential applications of cordycepin in other neurodegenerative diseases to remove neuropathological aggregates.

In addition to its functions in kidney and neuron diseases, cordycepin attenuated lung inflammation by decreasing pro-inflammation cytokines, including IL-1β and IL-6, in an LPS-induced acute lung injury mouse model [165]. Additionally, cordycepin suppressed the inflammasome signaling pathway in LPS-induced Raw 264.7 cells, in which autophagy proteins participate [166]. These shreds of evidence suggest that cordycepin may adjust immune responses through the regulation of the autophagy process. Thus, in future studies, strategies to maintain the balance of autophagy-associated immune pathways to prevent immune diseases require further investigations.

Most interesting, the immune system and the gut microbiota have a close relationship and regulate each other. The disturbance of intestinal microbes can seriously affect the physiological health of the host and cause many diseases. *Cordyceps militaris* and its functional components have beneficial effects on human health that are associated with the host gut microbiota. Extensive research on the extracellular polysaccharide of *Cordyceps militaris* has been carried out regarding its regulatory effect on gut microbiota [167]. It was also reported that cordycepin reduced the body weight and regulated the constitution of gut microbiota in high-fat diet (HFD)-induced obese rats [168]. In the study, cordycepin treatment reversed the relative abundance of Bacteroidetes and Firmicutes in the HFD-induced obese rats, resulting in similar phenotypes to the chow-fed diet group. However, further research is needed to better explain the complex interactions between cordycepin and the gut microbiota.

## 4. Cordycepin-Loaded Nanoparticles as a Promising Anti-Cancer and Health-Promoting Agent

Cordycepin is generally safe for human consumption [22]. However, several adverse effects, including dry mouth, nausea, abdominal distension, throat discomfort, headache, diarrhea, and allergic reactions have been reported after cordycepin treatment [19,169]. In addition, cordycepin is an adenosine analog; therefore, the metabolic and pharmacokinetic profiles of cordycepin resemble those of adenosine, which can be metabolized to 3′-deoxyinosine or undergo phosphorylation by adenosine kinase to convert into 3′-deoxyadenosine mono-, di-, and triphosphate. Nevertheless, cordycepin has a short half-life of 1.6 min, a high plasma clearance, low permeability, and high hepatic first-pass effects, leading to its low bioavailability in vivo [170]. Moreover, cordycepin is negatively charged and is thus rejected by cell membranes, leading to reduced cellular uptake [52]. Because of its toxicity and low bioavailability, potential novel drug delivery systems, such as nanoparticles, could be helpful to enhance the health-promoting effects of cordycepin. Bi et al. used a transferrin-conjugated liposome to deliver cordycepin to liver cancer cells [52]. This transferrin-conjugated liposome improved the solubility, biological activity, and anti-cancer effects of cordycepin [52]. Interestingly, Wu et al. indicated that cordycepin-loaded liposomes predominantly arrested liver cancer cells at the G2/M phase and induced higher levels of apoptosis than free cordycepin [171]. Another study conducted by Marslin et al. prepared cordycepin-loaded Poly(d,l-lactide-co-glycolide) (PLGA) nanoparticles (CPNPs) and compared their cellular uptake, cytotoxicity, and hemolytic potential with free cordycepin [172]. The results showed that CPNPs had better cytotoxicity effects, a longer half-life, and lower toxicity against breast cancer cells compared with free cordycepin [172]. Aramwit et al. incorporated cordycepin into the gelatin type A (GA) and gelatin type B (GB) nanoparticles and found that GA and GB nanoparticles had sustained release profiles [173], whereas GA nanoparticles showed higher anti-proliferative and anti-migratory effects on A549 lung cancer cells than GB nanoparticles [173]. In summary, these findings provide the important insight that using liposomes or nanoparticles would certainly be beneficial for the further development of novel cordycepin formulations to increase the biological effects, bioavailability, and half-life and reduce the toxicity of cordycepin. Therefore, the pharmacokinetic and toxicological profiling of these novel formulations of cordycepin should be carefully examined in the future.

## 5. Conclusions and Future Perspectives

*Cordyceps* is a natural medicinal mushroom that has been evaluated regarding its various biological activities, including anti-cancer, neuroprotective, and immune modulation effects and its renal protection activities. Recently, cordycepin has been recognized as a promising natural product for cancer chemoprevention because of its widespread and long history of use and lesser side effects. We reviewed the beneficial effect of cordycepin in cancers, neurodegenerative diseases, kidney damage, and the immune system through diverse mechanisms, including autophagy induction. Since autophagy is a very important pathway for regulating tissue homeostasis and may provide a potential therapy for many diseases, we suggest that future investigations of cordycepin should focus more on the mechanism of the autophagy pathway in vitro and in vivo to reveal its protective role in diseases. Moreover, because of the limitations of cordycepin used in vivo, another important issue is to modify the formulation of cordycepin to promote its bioactivity and bioavailability and lower its toxicity for further applications.

## Figures and Tables

**Figure 1 molecules-26-04954-f001:**
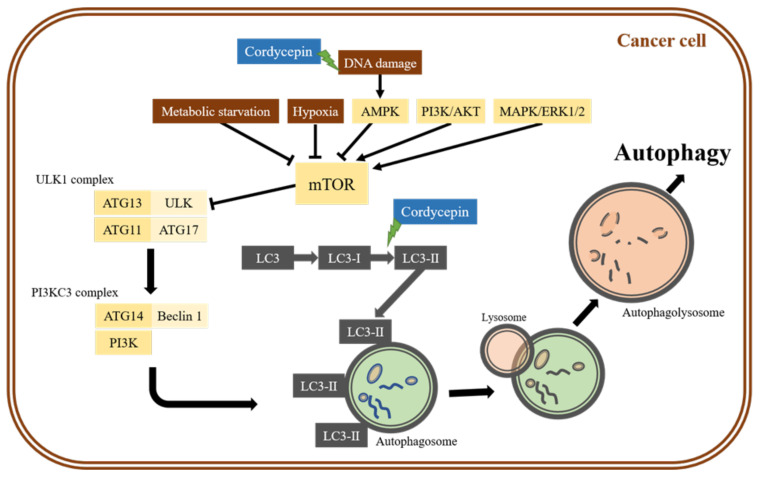
The anti-cancer mechanisms of cordycepin-induced autophagy. Various stress conditions, such as metabolic starvation, hypoxia, and DNA damage, initiate the autophagic process. Stress conditions inhibit mechanistic target of rapamycin (mTOR), leading to the activation of Unc-51-like kinase 1 (ULK1) complex, which triggers the formation of the phagophore by activating the “initiation complex”. Cordycepin could cause DNA damage and trigger the inhibition of the mTOR pathway to induce cancer cell death. The phosphatidylethanolamine-conjugated form of Microtubule-associated protein 1A/1B-light chain 3-II (LC3-II), which is converted from LC3-I by *ATG*4-dependent cleavage, is critically involved in the fusion of mature autophagosomes and lysosomes. The autophagolysosome, which is already fused with a lysosome, could then result in the degradation and recycling of metabolites. Cordycepin also upregulates LC3-II expression, leading to autophagy in cancer cells.

**Figure 2 molecules-26-04954-f002:**
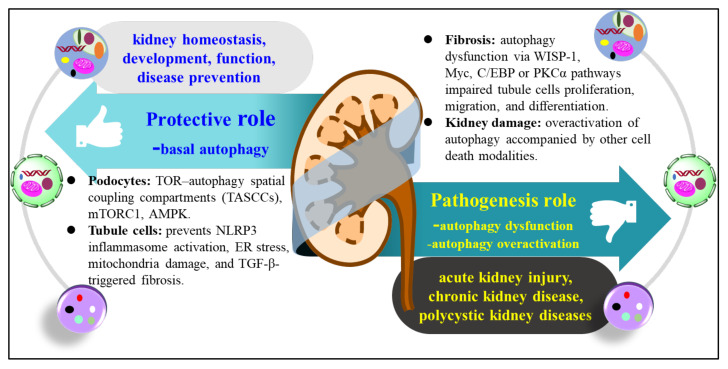
The dual role of autophagy in kidney homeostasis and disease. Basal autophagy is a protective role in the maintenance of kidney homeostasis, structure, and functions. TOR–autophagy spatial coupling compartments (TASCCs), mTORC1 and AMPK are major pathways contributed to the maintenance of podocyte homeostasis. In tubular cells, autophagy prevents inflammation and fibrosis via downregulating NOD-like receptor (NLR) family pyrin domain containing 3 (NLRP3) inflammasome activation and ER stress. However, pathogenesis role of autophagy contributes to acute kidney injury, chronic kidney diseases and polycystic kidney diseases via WNT family member 1 (WNT1)-induced signaling pathway protein-1 (WISP-1), Myc, CCAAT-enhancer-binding protein (C/EBP) or protein kinase C alpha (PKCα). Meanwhile, autophagy dysfunction or over activation accompanied by other cell death modalities might induce tubular atrophy, promote kidney fibrosis, thereby, aggravate kidney diseases.

**Figure 3 molecules-26-04954-f003:**
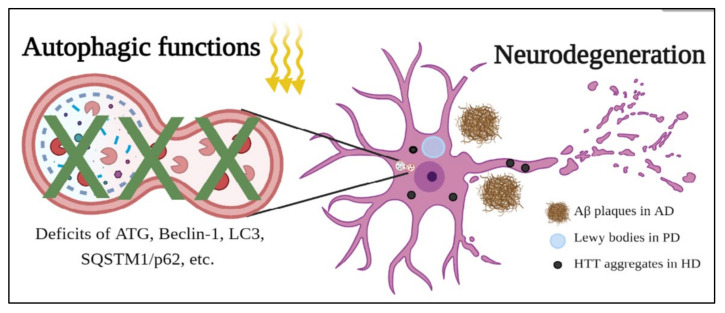
The role of autophagy in neurodegenerative diseases. The dysfunctions of autophagy are observed in neurodegenerative diseases, including AD, PD and HD, due to deficits of *ATG* family, Beclin-1, LC3, SQSTM1/p62, etc. Since the autophagy could not degrade disease-causing proteins properly, accumulations of Aβ plaques, Lewy bodies and HTT aggregates are detected in AD, PD and HD, respectively, finally leading to neurodegeneration.

**Figure 4 molecules-26-04954-f004:**
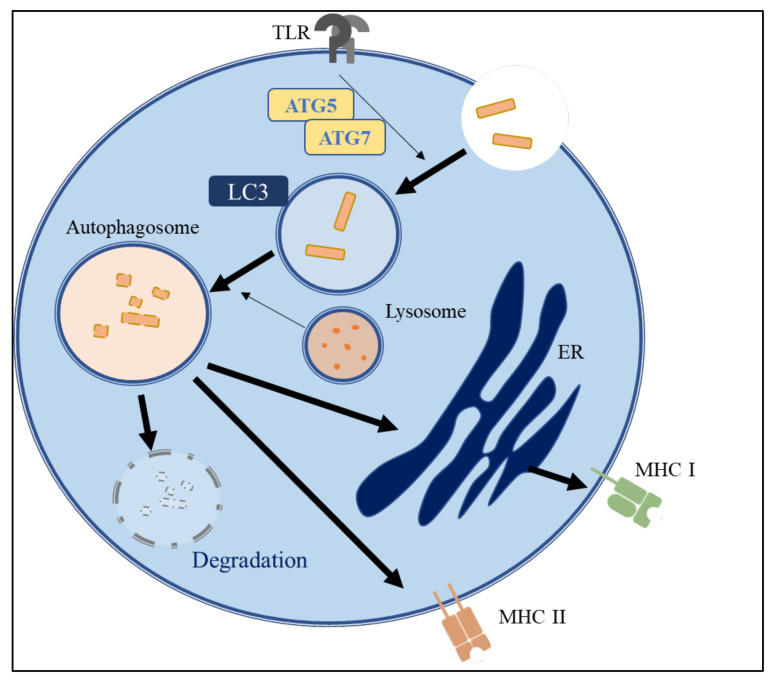
The role of autophagy in immune systems. Autophagy and autophagy-associated proteins are required for pathogen clearance and antigen-presenting signaling in the immune system. LC3 takes part in the capacity to clear apoptotic cells and pathogens in innate immune responses. In the adaptive immune system, ATG5 involves in MHC class I and MHC class II antigen presentation in APCs leading to the T cell responses. Autophagy-associated proteins plays an important role in the innate and adaptive immune regulation.

**Figure 5 molecules-26-04954-f005:**
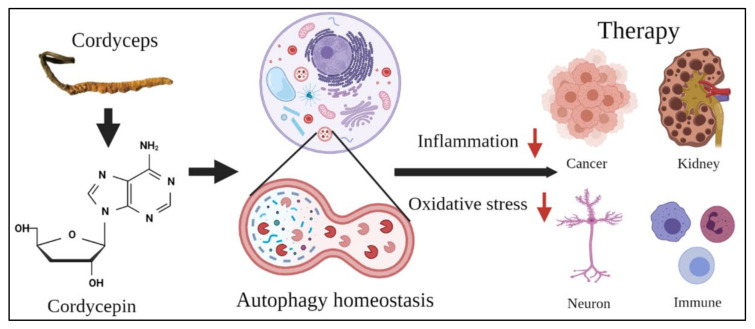
The possible potential therapy of cordycepin for different human diseases. The major extract cordycepin from *Cordyceps* regulates different stages of autophagy homeostasis inside cells to reduce inflammation and oxidative stress, providing further evidence of its potential as a candidate for therapy for cancer, kidney, neuron, and immune diseases.

## Data Availability

Not applicable.

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
