# Peer review of "The Role of Autophagy in Anti-Cancer and Health Promoting Effects of Cordycepin"

_molecules, 2021, doi:10.3390/molecules26164954_

Round 1
Reviewer 1 Report
This is a very well-written review about the role of autophagy in anticancer and health-promoting effects of Cordycepin. The authors address the anti-cancer mechanisms of Cordycepin including EGFR, MAPKs and GSK-3β, leading to cell cycle arrest or apoptosis. Furthermore, Cordycepin induces autophagy to trigger cell death, inhibits tumor metastasis and modulates immune systems. In addition, dysregulation of autophagy is associated with cancers and other diseases. The authors think that formulation changes for Cordycepin enhance its bioactivity and bioavailability, and lower toxicity for future applications. Therefore, the understanding of the autophagy mechanism provides a novel mechanistic insight of anti-cancer and health-promoting effects by Cordycepin. This review is of high relevance and a pleasure to read. However, there are still several minor points that need to be addressed.
- Page 3, line 102, there is one mistake of punctuation. Also, some mistakes of punctuation have appeared in the Please carefully check these punctuations.
- Page 3, line 128, “such as Autophagy Related (ATG) family” should be changed to “such as autophagy-related genes (ATG) family”.
- Spell out the abbreviation of LC3 and PLGA when they first time appear in the manuscript.
- I suggest that the authors provide the schematic figures of autophagic effects in several diseases in section 3 “The role of autophagy in health-promoting effects of Cordycepin” (3.1~3.3).
- The grammar and syntax are not correct in some places in this manuscript. It would be helpful to have the manuscript carefully reviewed and corrected by an author or colleague who is a primary English speaker.
Author Response
Reviewer 1
This is a very well-written review about the role of autophagy in anticancer and health-promoting effects of Cordycepin. The authors address the anti-cancer mechanisms of Cordycepin including EGFR, MAPKs and GSK-3β, leading to cell cycle arrest or apoptosis. Furthermore, Cordycepin induces autophagy to trigger cell death, inhibits tumor metastasis and modulates immune systems. In addition, dysregulation of autophagy is associated with cancers and other diseases. The authors think that formulation changes for Cordycepin enhance its bioactivity and bioavailability, and lower toxicity for future applications. Therefore, the understanding of the autophagy mechanism provides a novel mechanistic insight of anti-cancer and health-promoting effects by Cordycepin. This review is of high relevance and a pleasure to read. However, there are still several minor points that need to be addressed.
Response:
We thank to reviewer’s appreciation and suggestions for our manuscript.
Comment:
Page 3, line 102, there is one mistake of punctuation. Also, some mistakes of punctuation have appeared in the Please carefully check these punctuations.
Response:
Thank you for pointing out the errors. We have corrected the punctuation in all content accordingly. Thank you.
Comment:
Page 3, line 128, “such as Autophagy Related (ATG) family” should be changed to “such as autophagy-related genes (ATG) family”.
Response:
Thank you for the correction. We have changed the full name of Autophagy Related (ATG) family to autophagy-related genes (ATG) family in line 128.
Comment:
Spell out the abbreviation of LC3 and PLGA when they first time appear in the manuscript.
Response:
Thank you for the correction. We have added the full name of LC3 and PLGA in line 148, 157 and line 440.
Comment:
I suggest that the authors provide the schematic figures of autophagic effects in several diseases in section 3 “The role of autophagy in health-promoting effects of Cordycepin” (3.1~3.3).
Response:
Thank you for your suggestion, we have added the figures of autophagic effects in different diseases in section 3.1 (Figure 2), 3.2 (Figure 3) and 3.3 (Figure 4).
Comment:
The grammar and syntax are not correct in some places in this manuscript. It would be helpful to have the manuscript carefully reviewed and corrected by an author or colleague who is a primary English speaker.
Response:
Thank you for the comment. We have checked the manuscript and corrected all the incorrect grammar and syntax in the manuscript
.
Reviewer 2 Report
Review
The Role of Autophagy in Anticancer and Health Promoting Effects of Cordycepin,
by Chen et al.
This review analyzes the role that an adenosine nucleoside, cordycepin, present and isolated from the entomopathogenic fungus Cordyceps sinensis, in the process of autophagy, both in cancer situations and as a promoter of health improvement.
This is a review carried out in some depth, the main characteristic and novelty of which is to analyze the molecular mechanisms, which have recently been shown, and which are involved in the induction of autophagy by cordycepin, as well as its positive impact as an anticancer and health promoter.
The work is well organized, it deals with the main aspects that should be included in a review in an orderly manner, it is very current and it is written in such a way that the reader is caught in their reading. In my case, I have to confess that I have enjoyed both reading it and the content.
Another positive value of this review are its figures since they are authentically illuminating in what is intended to communicate, however, I believe that the authors should include some more, even if they are illustrative diagrams, in other sections that are discussed in the work, since, on those occasions, readers who are further away from the topic may not understand the message clearly.
For all these reasons, I believe that this work should be accepted as long as the authors include those diagrams that I comment, specifically those related to sections 3.1, 3.2 and 3.3.
Author Response
Reviewer 2.
The Role of Autophagy in Anticancer and Health Promoting Effects of Cordycepin,
by Chen et al.
This review analyzes the role that an adenosine nucleoside, cordycepin, present and isolated from the entomopathogenic fungus Cordyceps sinensis, in the process of autophagy, both in cancer situations and as a promoter of health improvement.
This is a review carried out in some depth, the main characteristic and novelty of which is to analyze the molecular mechanisms, which have recently been shown, and which are involved in the induction of autophagy by cordycepin, as well as its positive impact as an anticancer and health promoter.
The work is well organized, it deals with the main aspects that should be included in a review in an orderly manner, it is very current and it is written in such a way that the reader is caught in their reading. In my case, I have to confess that I have enjoyed both reading it and the content.
Another positive value of this review are its figures since they are authentically illuminating in what is intended to communicate, however, I believe that the authors should include some more, even if they are illustrative diagrams, in other sections that are discussed in the work, since, on those occasions, readers who are further away from the topic may not understand the message clearly.
Response:
We appreciate the reviewer’s highly compliments. Our response to reviewer’s suggestion is shown as follows.
Comment:
For all these reasons, I believe that this work should be accepted as long as the authors include those diagrams that I comment, specifically those related to sections 3.1, 3.2 and 3.3
Response:
Thank you for your suggestion, we have added the Figure 2, 3 and 4 of the autophagic effects in different diseases in section 3.1, 3.2 and 3.3.